# Comparison of Estonian Schoolchildren’s Physical Activity and Fitness Indicators before and after the COVID-19 Pandemic’s Period of Restricted Mobility

**DOI:** 10.3390/ijerph21060744

**Published:** 2024-06-06

**Authors:** Saima Kuu, Joe Noormets, Triin Rääsk, Kirsti Pedak, Viktor Saaron, Karin Baskin, Kristjan Port

**Affiliations:** School of Natural Sciences and Health, Tallinn University, 10120 Tallinn, Estonia; joe.noormets@tlu.ee (J.N.); triin.raask@tlu.ee (T.R.); kirsti.pedak@tlu.ee (K.P.); viktor.saaron@tlu.ee (V.S.); kristjan.port@tlu.ee (K.P.)

**Keywords:** physical activity, physical fitness, schoolchildren, COVID-19 pandemic, restricted mobility

## Abstract

It is recognized that the time adolescents spend on physical activity, and the corresponding physical fitness indicators, have diminished over time. However, the exact impact of the COVID-19 pandemic restriction period on physical activity and health-related physical fitness indicators remains unclear. This study sought to determine if and to what degree the primary indicators of physical activity (exercise frequency, exercise intensity, and outdoor physical activity) and health-related physical fitness (strength and endurance) among schoolchildren have shifted, by comparing data from before and after the coronavirus pandemic period. Students aged 12–17 years took part in the study. The physical activity questionnaire and fitness tests were conducted in the autumn of 2017 and spring of 2022. The main results show that moderate- and vigorous-intensity physical activity time and time spent actively outdoors have decreased among adolescents (*p* < 0.05). There were no significant changes in exercise frequency (*p* > 0.05). However, there was a statistically significant decline in strength (standing broad jump, bent arm hang) (*p* < 0.05) and endurance (20 m shuttle run, curl-up) (*p* < 0.01) when comparing results from before and after the COVID-19 pandemic restriction period. In conclusion, restricted mobility had the greatest impact on the time spent outdoors and, in turn, on physical fitness.

## 1. Introduction

Increasing concerns have heightened the focus on children’s physical activity and fitness. In a systematic review, Masanovic and colleagues (2020) highlighted changes in the physical fitness of schoolchildren over the past four decades, noting that indicators of strength and endurance have consistently declined over time [1]. Europe has set reference standards for evaluating physical fitness [2,3] and offers a service that supports monitoring, providing pan-European comparative data—FitBack [4]. It is well known that to improve physical fitness, children aged 5–17 must engage in moderate to vigorous physical activities, as recommended by the WHO [5]. Both regular weekly training [6,7] and short, intense exercises during school [8] contribute to the development of strength and endurance.

It is widely recognized that a child’s physical fitness level impacts their health in adulthood, with strength and endurance indicators being particularly crucial [9,10]. Comparable links can also be drawn with physical activity. Through their longitudinal study, Kemper and Monyeki (2019) demonstrate “that the promotion of physical activity (including physical education and sport) in adolescence can potentially be a strong tool to prevent chronic diseases and reduce healthcare costs later in life” [11].

It is encouraging to observe that, according to the HBSC (Health Behaviour in School-aged Children) study, the physical activity levels among school children have been relatively stable in recent years [12,13]; meanwhile, boys are more likely to meet the recommended daily physical activity levels than girls, and younger school students (11 years old) are more physically active than older ones (15 years old) [12,13,14]. Additionally, over the past two decades, leisure time physical activity has declined among both rural and urban children, with a particularly significant decrease in the time rural children spend outdoors [15]. At the same time, the sedentary screen time for both rural and urban children has increased [13,15,16], and this in turn limits physical activity [17]. Excessive screen time and reduced physical activity contribute to weight gain [18] and various health issues like back pain [16] and depressive symptoms [19].

Like the rest of the world, Estonia was also “closed”. In the spring of 2020, at the end of March, the government of the Republic of Estonia mandated that all school children engage in distance learning, which naturally led to movement restrictions, the suspension of organized training, and the elimination of active commuting, such as traveling between school and home.

Several studies show a decrease in the proportion of physical activity in 2020, when the lockdown period was introduced due to the COVID-19 pandemic, compared to the period before the pandemic [20,21]. Rossi and colleagues (2021) summarized findings from 75 different studies in a scoping review, highlighting that the majority of the analysed studies (57) reported a decline in both the duration and frequency of physical activity among children and adolescents during the pandemic; on average, the time spent on physical activity per day decreased by 68 min [22]. Neville et al. (2022) observed that total daily physical activity in children and adolescents decreased by 20% during the lockdown period due to restrictions requiring physical distancing [23]. The decline in the physical activity of schoolchildren around the world also led to a decrease in indicators of physical fitness, especially changes in strength and endurance [24,25]. Drawing on multiple surveys, Knight and colleagues (2022) observed an increase in the overall sedentary time among children and adolescents during the COVID-19 pandemic, particularly in older children [26]. Physically inactive time mainly comes at the expense of increased screen time. The number of students with more than 2 h of screen time per day increased by 22%, from 66% to 88% [26]. However, the impact of age and gender on physical activity levels remains uncertain [26].

Two years after life largely returned to normal, the question remains: have there been changes in the physical activity of schoolchildren compared to the pre-pandemic period, and what impact have the restrictions had on the physical fitness of the new generation of teenagers? Prompted by this inquiry, this study seeks to determine the extent to which the primary indicators of physical activity and health-related physical fitness have changed among schoolchildren, focusing specifically on the Estonian school-aged population (12–17 years). Based on the purpose of this work, the following research questions were established: (a) How have the exercise frequency, exercise intensity, and outdoor physical activity of schoolchildren changed when comparing these indicators before and after the period of COVID-19 restrictions? (b) How have the strength and endurance of schoolchildren changed when comparing these indicators before and after the period of COVID-19 restrictions?

## 2. Materials and Methods

Background of the database: In 2017, we conducted “The Study of Testing the Physical Fitness and the Physical Activity of Estonian Schoolchildren and the Factors That Affect It”. This research aimed to gather comprehensive data on the physical fitness and activity levels of schoolchildren, marking the first large-scale assessment of Estonian schoolchildren’s physical fitness since the previous century [27]. During the autumn semester of 2017, children from the 6th, 8th, and 10th grades were selected for examination based on a random sample of schools. A 2018 publication summarizing the study [28] revealed that a significant number of 6th-grade boys were categorized as being in the health-risk group due to their physical fitness levels, according to age-appropriate European standards [2]. Based on this, a new goal to conduct a follow-up survey after four years was established to observe how the 6th-grade indicators changed for new students and to determine whether weak results in the 6th grade persisted into the 10th grade. The follow-up study was conducted in the 2021/2022 academic year (hereinafter 2022, as the data were collected mainly in the spring semester) and it coincided with the post-COVID-19 period, which in turn affected the collection of results. The study included the same age groups, but not the same subjects. In 2022, assessments were conducted with new students who had reached the 6th, 8th, and 10th grades.

### 2.1. Participants

The participants (12–17-year-olds) were 6th, 8th, and 10th-grade students (henceforth accordingly 12–13-y, 14–15-y, and 16–17-y groups). In 2017, a total of 3419 students, of whom 1691 were girls and 1722 were boys, and in 2022, 3034 students, of whom 1505 were girls and 1529 were boys, took part in the Estonian school youth physical activity and fitness study. Participation in the study was voluntary, and in the spring of 2022, intermittent self-isolation among school students due to the spread of COVID-19 resulted in daily fluctuations in the number of participants. The list below shows the number of participants and the average age (M ± SD years) according to age groups and genders that participated in the study during the different testing years.

2017 females:12–13-y—n = 643, age: 12.15 ± 0.3614–15-y—n = 579, age: 14.21 ± 0.4116–17-y—n = 469, age: 16.15 ± 0.362017 males:12–13-y—n = 694, age: 12.20 ± 0.4014–15-y—n = 581, age: 14.25 ± 0.4416–17-y—n = 447, age: 16.21 ± 0.412022 females:12–13-y—n = 631, age: 12.29 ± 0.4614–15-y—n = 514, age: 14.28 ± 0.4516–17-y—n = 360, age: 16.28 ± 0.452022 males:12–13-y—n = 667, age: 12.36 ± 0.4814–15-y—n = 564, age: 14.33 ± 0.4716–17-y—n = 298, age: 16.28 ± 0.48

### 2.2. Organization of the Study

The 2017 study included 43 randomly selected schools, while in 2022, 25 schools participated. Both studies were conducted among students in the 6th, 8th, and 10th grades.

Before conducting the study, consent was asked from the school management. The entire study was then introduced to the selected classes, and the informed consent of the students was also asked so that they could participate in the study. Each student personally gave their informed consent to the class teacher. Information about the study was also sent to the parents through the class teacher, and so-called passive consent was expected from them. The study information was sent to the children’s parents twice, three weeks apart, to ensure that they received it. Parents who agreed to their child’s participation did not need to respond to the information letter. If the parent disagreed with their child’s participation in the study, they were required to notify the study organizer.

The questions related to physical activity were distributed via Google Forms, and students completed this self-reported questionnaire either during their class teacher’s session or in the computer education class. Each class was accompanied by a specially trained researcher who could answer any questions and assist students in understanding the questionnaire.

Physical fitness assessments were conducted during physical education classes. Subjects completed a general warm-up before the strength and endurance testing. The tests were conducted by lecturers (most of the authors of this article) and students on the study team, all of whom had received identical instructions and training. The testing conditions in the academic years 2017/2018 and 2021/2022 were consistent. The physical fitness tests took place indoors in the sports hall of the participating school, eliminating the influence of environment novelty and possible weather conditions on the test results. Participation in the study was entirely voluntary, and students could withdraw at any time without providing a reason.

During the 2017/2018 academic year, the tests and questionnaires were primarily conducted in the autumn semester, from October to December. Therefore, the results are presented for the year 2017. Similarly, during the 2021/2022 academic year, the tests and questionnaires were primarily conducted in the spring semester, from February to May. Therefore, the results are presented for the year 2022.

The study mentioned above received approval (application no 6-5.1/34) from the Ethics Committee of Tallinn University.

### 2.3. Methods

#### 2.3.1. Physical Activity Questionnaire

The authors of this study developed a self-reported physical activity questionnaire. This study evaluates exercise frequency, hours of intensive exercise, and time spent physically active outdoors as indicators of physical activity.

The frequency of exercise was determined using the following question:

Outside of school hours: How often do you usually exercise (at least 20 min at a time)?

Never;Less than once a month;Once a month;Once a week;2–3 times a week;4–6 times a week;Every day.

For data analysis, the responses were grouped into three groups:“low frequency”—answer options 1–4;“moderate frequency”—answer option 5;“high frequency”—answer options 6–7.

The hours of intensive exercise were determined using the following question:

Outside of school hours: How many hours a week do you usually exercise so that you pant and sweat?

8.None;9.About 0.5 h;10.About 1 h;11.About 2–3 h;12.About 4–6 h;13.7 h or more.

For data analysis, the responses were grouped into three groups:“low intensity hours”—answer options 1–3;“moderate intensity hours”—answer option 4;“high intensity hours”—answer options 5–6.

The physically active time outdoors was determined using the following question:

How many hours a week are you physically active outside in your free time (do you walk, run, play ball with friends, ride a bicycle or skateboard; walk your dog, etc.?)

14.None;15.About 0.5 h;16.About 1 h;17.About 2–3 h;18.About 4–6 h;19.7 h or more.

For data analysis, the responses were grouped into three groups:“low active outdoor hours”—answer options 1–3;“moderate active outdoor hours”—answer option 4;“high active outdoor hours”—answer options 5–6.

The objectivity of the answers to the questionnaire was confirmed by statistically significant correlation coefficients between these characteristics (Spearman’s rho = 0.218–0.584, *p* < 0.001).

#### 2.3.2. Physical Fitness Tests

The study primarily concentrated on health-related physical fitness, incorporating measures of strength and endurance. For strength indicators, leg explosive strength was assessed using the standing broad jump test, and arm functional strength through the bent-arm hang test. For endurance indicators, trunk muscle endurance was evaluated with the curl-up test, and cardiorespiratory system endurance with the 20 m shuttle run test. The tests were performed under the conditions described below:

In the standing broad jump test, the student stands shoulder-width apart, toes behind the starting line, hands in front parallel to the ground. As the student bends their knees, they vigorously swing their hands backward. During the push-off, the student swings their hands forward and upward, then jumps as far as possible. The student must definitely push off with the feet together. The student must land on two feet side by side in a balanced position so that no other part of the body touches the ground after landing. The result is read from the measuring tape or according to the scale marked on the mat and is measured in meters with an accuracy of 1 cm. The test is performed twice in a row, and the best result counts [29].

In the bent-arm hang test, the student hangs his chin over the bar for the maximum length of time. The hands must be in an overhand grip and the arms are bent, with the chin over the bar but not touching it. A bench is used to get into the measuring position. The time starts when the student’s feet no longer touch the bench and time is stopped when the student can no longer maintain the correct position. Each student has one attempt, and the result is measured in seconds [29].

In the curl-up test, the student lies on his back on the mat, hands next to the body, knees bent 90 degrees, and heels and soles against the mat. The student must sit up so that the elbows also come off the mat and the fingers move along the mat, 10 cm from the line marked on the mat. The student repeats shoulder girdle pull-ups as many times as possible, with a rhythm of 25 times per minute for 3 min in a row. The test is only performed one time and the result is the number of correctly performed attempts, with a maximum of 75 repetitions [30].

In the 20 m shuttle run test, the student must run from line to line along a 20 m track. The initial speed is 8.5 km/h, which accelerates by 0.5 km/h every minute. The student must run to the rhythm of the sound signal and run as long as he can. The student must reach the finish line each time before the sound signal. There is a warning area of 2 m on both sides before the finish line, where the student must definitely reach before the next sound signal. If the student does not reach the finish line before the sound signal twice in a row, the test is over for him. If a student does not reach the finish line one time but does it again the next time, they can continue running. The test is performed one time and the result is the number of line sections passed [29].

All the physical fitness tests in this study are highly repeatable, validated, and reliable [31,32], and they belong to the HELENA (Healthy Lifestyle in Europe by Nutrition in Adolescence) Study [33].

### 2.4. Statistical Analysis

IBM SPSS Statistics 25 was used for the statistical analysis. Descriptive statistics (n, means, standard deviations, 95% confidence intervals) were utilized to characterize the anthropometric indicators and physical fitness test results of the observed groups. Frequency distribution was employed to present the physical activity indicator results, which are shown as percentages.

The chi-square test was used to compare the frequency distributions of physical activity indicators across different academic years. Differences in physical fitness indicators between the academic years were assessed using a univariate ANOVA, along with the Bonferroni post hoc test. The level of significance was set at *p* < 0.05. The univariate ANOVA test was employed to check the main and interaction effects of the factors: year of testing, age, and gender. When factors influenced the results, the F-statistic, degrees of freedom (df), and partial eta squared (η^2^) characterizing the effect size were presented.

## 3. Results

### 3.1. Descripitive Results

Descriptive statistics (n, means, standard deviations, 95% confidence intervals) for participants’ anthropometric data for both study years are presented in Table 1.

### 3.2. Physical Activity

There were no statistically significant differences (*p* > 0.05) in exercise frequency between the pre-pandemic and post-pandemic periods (Table 2). Participants across all age and gender groups were similarly distributed based on exercise frequency. Those who do not exercise or do so very infrequently (up to once a week) were in the minority. The groups exercising with moderate and high frequency were nearly balanced, though there were slightly more participants who engaged in training 4–7 times a week. However, when examining the changes in exercise frequency by age, it was observed that in 2017, the frequency of exercise among 16–17-year-old girls was significantly lower (χ^2^ = 17.589, *p* = 0.001) compared to other age groups. In 2022, as girls aged, the frequency of their sports participation did not show significant changes (χ^2^ = 1.962, *p* = 0.743). The study found no statistically significant differences in exercise frequency among boys as they aged, both in 2017 and in subsequent observations (χ^2^ = 6.120, *p* = 0.190) in 2022 (χ^2^ = 3.563, *p* = 0.468).

Comparing the 2022 data on intensive exercise time with the results from 2017 (Table 3), it can be seen that “moderate intensity hours” significantly decreased after the pandemic period, and “low-intensity hours” significantly increased (*p* < 0.01). However, the changes in the proportion of exercise intensity hours during the week in most gender and age groups were not statistically significant (*p* > 0.05), except for 12–13-year-old girls, among whom the proportion of “low-intensity hours” increased remarkably and the proportion of “high-intensity hours” decreased (*p* < 0.01) during the pandemic period. The comparison between age groups also did not reveal statistically significant differences in the time of intensive exercise. The results of the Chi-Square tests were as follows: girls in 2017—χ^2^ = 5.906, *p* = 0.206; girls in 2022—χ^2^ = 5.030, *p* = 0.284; boys in 2017—χ^2^ = 2.928, *p* = 0.570; boys in 2022—χ^2^ = 4.410, *p* = 0.353.

Among the indicators of physical activity, the time spent being physically active outdoors saw the most significant changes during the pandemic period (Table 4). Overall, comparing the 2022 results with those from 2017 across all observed age and gender groups, there was an increase in the proportion of “low active outdoor hours” and a decrease in “high active outdoor hours”. These changes were statistically significant particularly among 12–13-year-old (*p* < 0.001) and 14–15-year-old girls (*p* < 0.001) and among 12–13-year-old (*p* < 0.05) and 16–17-year-old boys (*p* < 0.01). The comparison between age groups did not reveal statistically significant differences in the physically active time outdoors. The results of the Chi-Square tests were as follows: girls in 2017—χ^2^ = 6.422, *p* = 0.107; girls in 2022—χ^2^ = 2.514, *p* = 0.642; boys in 2017—χ^2^ = 3.452, *p* = 0.485; boys in 2022—χ^2^ = 7.488, *p* = 0.112.

### 3.3. Physical Fitness

The results of the standing broad jump (Table 5) demonstrated statistically significant changes across all age and gender groups during the pandemic period (*p* < 0.05). For girls of all ages and boys aged 14–17 years, the results were significantly lower (*p* < 0.05). However, among boys aged 12–13 years, there was a slight statistically significant improvement (*p* < 0.05).

The detailed analysis (ANOVA) of the separate factors revealed a year of testing interaction effect for age (F (2) = 479.749, *p* < 0.001, partial η^2^ = 0.157) and gender (F (1) = 1680.435, *p* < 0.001, partial η^2^ = 0.246), highlighting that the test proficiency improved in both genders with age.

The results of the bent-arm hang test, measured in time (Table 5), statistically significantly worsened (*p* < 0.05) across all observed gender and age groups during the pandemic period, with the exception of 12–13-year-old boys, for whom there was no significant effect. The effect of year of testing on gender (F (1) = 994.558, *p* < 0.001, partial η^2^ = 0.164) and age (F (2) = 157.973, *p* < 0.001, partial η^2^ = 0.059) demonstrated a significant effect, showing that the results of the boys’ bent-arm hangs improved with increasing age, while the girls’ bent-arm hang time decreased with increasing age (Table 5), independent of per- or post-pandemic stage.

During the pandemic period, the results of both endurance tests, the curl-up test and the 20 m shuttle run test (Table 6), showed a statistically significant decline across all gender and age groups (*p* < 0.01).

In the curl-up test, there was a small but statistically significant effect of the year of testing (F (1) = 91.699, *p* < 0.001, partial η^2^ = 0.018) and gender (F (1) = 126.359, *p* < 0.001, partial η^2^ = 0.024). Boys showed improvement with age, whereas for girls, age tended to have a slight-negative or no effect on performance. However, both genders experienced a decline in performance in the post-pandemic measurement (Table 6).

The 20 m shuttle run, which tests the endurance of the cardiovascular and respiratory systems, showed statistically significant moderate-to-large effects of the year of testing (F (1) = 196.697, *p* < 0.001, partial η^2^ = 0.038), age (F (2) = 154.627, *p* < 0.001, partial η^2^ = 0.059) and gender (F (1) = 764.614, *p* < 0.001, partial η^2^ = 0.135). Boys’ performance improved with age, whereas this improvement was less evident among girls. Both genders experienced a decline in performance in the post-pandemic measurement (Table 6).

## 4. Discussion

The findings of this study, which compares indicators of physical activity and fitness before and after the COVID-19 pandemic, reveal significant changes in the amount of time schoolchildren spend being physically active outdoors (with a reduction in weekly hours) and a decline in both strength and endurance.

This study observed that the frequency of exercise among Estonian schoolchildren remained largely unchanged when comparing the academic years of 2021 and 2017, with nearly 80% of students engaging in training 2–7 times a week. Moreover, more than 40% of them exercise regularly 4–7 times a week. This study’s results agree with the HBSC survey of Estonian students, where 41–42% of students were moderately physically active for at least 1 h at a time on five or more days a week in 2017 and 2022, respectively, [13]. There is also a gender difference similar to the results of the HBSC study, where boys perform a higher frequency of exercise than girls [13]. The results of the German HBSC study also show a higher sports activity in boys compared to girls, where almost 50% of boys do sports four or more times a week, and about 30% of girls are equally active [12]. Looking back at the time of the COVID-19 restrictions, it seems that the lockdown period did not affect the frequency of sports activity. For example, in April 2020, 41.8% of Canadian youth aged 14–18 years were moderately or vigorously physically active 4–7 times a week [34]. However, both Canadian and US researchers found that time spent on physical activity decreased during the pandemic period [20,21].

While this study found no significant changes in overall exercise frequency during the pandemic, it is worth noting that exercise frequency (4–7 times a week) increased in most age and gender groups, with the exception of 12–13-year-old girls. Additionally, exercise frequency tended to decrease with age, though this trend was only statistically significant among girls who participated in the study in 2017. The percentage of girls who rarely engage in regular exercise increased significantly with age. While 22.2% of 12–13-year-olds fall into the category of non-trainers, this figure rises to 31.5% among 16–17-year-olds. The Estonian HBSC study also confirms that the frequency of sports among both boys and girls decreases with age [13]. According to the Estonian HBSC study, 22% of 11-year-olds meet the physical activity recommendations, compared to just 12% of 15-year-olds [13]. Similarly, Bucksch et al. (2024) report a comparable age-related decline in exercise frequency among German adolescents [12]. In their systematic review, Batista et al. (2019) points out that a greater frequency of sports in childhood is related to higher physical activity in adulthood [35], and that today’s sports policies in public health should also be based on this. In addition to the above, it is important to point out that significantly less anxiety and depression symptoms occur among adolescents involved in sports than among adolescents who do not participate in sports [36]. Additionally, Cheon and his colleagues (2021) have shown that exercise frequency is associated with greater sociality and happiness [37].

This study found no notable changes in the duration of intensive exercise in terms of overall sports frequency. While the general data analysis indicated a decrease in the weekly amount of exercise that induces sweating and heavy breathing among schoolchildren from the pre- to the post-pandemic period, significant shifts were absent across separate gender and age groups. The exception was 12–13-year-old girls, among whom there was an increase of more than 10% in “low intensity hours”. The findings of our study closely align with those of a study conducted in Japan, which revealed that exercise time among 5th and 8th graders decreased by an average of 10% during the COVID-19 pandemic. The reduction was smaller among younger students (5.8%) and larger among older students (14.2%) [25]. A study conducted on 22-year-old Italian athletes also showed the same tendency, as the proportion of high-intensity physical activity decreased during the coronavirus period [38]. And, Rossi et al. (2021) confirmed the fact that according to most studies, the duration and frequency of physical activity among children and adolescents decreased during the period of the coronavirus pandemic, and other studies have indicated that the time spent on physical activity per day decreased by at least 45 min and up to 91 min. [22]. But, at the same time, the change in the proportion of time spent playing sports during the COVID-19 restrictions depended on the sport. A study conducted in Australia clearly shows that the time spent practicing outdoor sports, which are also more individual sports, including running, jogging, and cycling increased, while the time spent practicing team sports, mostly played indoors, decreased [39]. The time spent on moderate- and high-intensity physical activity is of decisive importance for the intensity of sports among schoolchildren and adolescents. The World Health Organization recommends that children aged 5–17 years engage in 60 min of moderate to vigorous daily physical activity for their health [5]. Tapia-Serrano et al. (2023) point out in their article that those who fulfil the 24 h physical activity guidelines of the WHO show better indicators of cardiovascular capacity in the 20 m shuttle-run test and in the standing long jump test, which support the health of the musculoskeletal system [40].

Being physically active outdoors has a positive effect on well-being [41]. The time adolescents spent being physically active outdoors decreased significantly during the pandemic period. In this study, “high active outdoor hours” decreased and the “low active outdoor hours” increased in all observed groups. The smallest changes occurred among 16–17-year-old girls and 14–15-year-old boys, where no significant shift in the reduction of hours spent outdoors was observed. Nigg et al. (2022) also noted a decrease in the time spent outdoors among adolescents and, at the same time, an increase in screen time [15]. Distance learning during the restrictions caused by the COVID-19 pandemic also contributed to the increase in screen time and, therefore, to the decrease in outdoor time. This was also shown by Knight et al. (2022), who, on the basis of various studies, showed that during the pandemic, the screen time and general sedentary time of children and adolescents increased [26]. Longer screen time, in turn, is mainly associated with an increase in symptoms of mental health disorders among adolescents [19]. The WHO report also points out that mental health problems and decreased physical activity were the two main health-related indicators associated with the period of the COVID-19 pandemic [42]. The main reason that the time school children spent outdoors decreased is probably related to social changes during the coronavirus period. In addition to closing schools and sports clubs, outdoor playgrounds and public training grounds were also closed.

In addition to the decrease in moderate and vigorous physical activity (MVPA) time and time spent outdoors, it is important to note that health-related physical fitness indicators also significantly deteriorated among adolescents. This study’s comparison of strength and endurance indicators before and after COVID-19 restrictions reveals a statistically significant decline in physical fitness across all genders and age groups. The only exception among all the evaluated gender and age groups was the 12–13-year-olds in terms of strength indicators, as their standing broad jump results were significantly even better in 2022 than in 2017 and their bent-arm hang test results did not change significantly over time. Jarnig et al. (2022) conducted a study evaluating the impact of COVID-19 on younger schoolchildren and obtained similar results—cardiorespiratory endurance indicators decreased more significantly than muscle strength indicators in 6–10-year-old children [43]. In their study, Wahl-Alexander et al. (2021) demonstrated that during the pandemic, the strength and endurance indicators of 8th grade students significantly declined [24]. Similarly, Kidokora et al. (2023), who examined millions of Japanese children, reported that physical fitness, particularly cardiorespiratory endurance measured by the 20 m shuttle run test, as in our study, deteriorated over time for 5th and 8th grade students [25].

Despite the fact that in the last decade there has been a lot of talk about the importance of physical fitness and fitness testing in physical education classes at school [4,44] and that fact that norms have been developed by which students can assess their fitness level [1,2,3], the physical fitness of schoolchildren is still in a downward trend. In other words, awareness of the problem has not stopped the decline in the level of physical fitness of children and adolescents. And we know that this also affects the health of schoolchildren in every way [9,10]. It has been found that assessment of the quality of life is related to cardiorespiratory endurance even in the case of 10-year-old schoolchildren [45]. And we know that playing sports in childhood and adolescence ensures better physical fitness as an adult [46].

What can we do? Fortunately, the situation can still be improved. Huhtiniemi and colleagues (2023) have shown that if physical fitness exercises are consistently integrated into school programs, all indicators of health-related physical fitness improve [8]. And, of course, regular weekly exercise helps to increase the level of physical fitness, especially just muscle strength and cardiorespiratory endurance [6]. And, physical fitness, in turn, is related to cognitive function and working memory [7] and also generally to academic success and quality of life [47]. Ericsson (2020) addresses both of the aspects mentioned above, noting that increasing the number of physical education classes and incorporating more motor skill exercises can enhance students’ physical fitness, which in turn improves their academic performance [48]. In 2016, Estonia launched the “A school that invites to move” initiative, allowing participating schools to acquire various tools through the project to enhance students’ physical activity and make the school day more active [49]. Despite these efforts, active breaks at school still fall short of helping students meet the daily recommended physical activity norms [50]. In summary, despite the aid programs, the physical fitness of schoolchildren has continued to decline over the years.

### Limitations of the Study

During the follow-up study, the COVID-19 virus was still spreading, resulting in many student absences during the physical activity and fitness assessments due to self-isolation from illness.

The tests were conducted in different seasons of the year. The data from 2017 were primarily collected in November, a month less conducive to outdoor activities due to unfavourable weather, which led to reduced outdoor time for everyone. In contrast, the data from 2022 were gathered in April–May, during the onset of spring in Estonia, a time when young people are more inclined to be outdoors.

Additionally, the tests were not conducted by the same individuals in all the participating schools, although everyone received the same instructions and training for conducting the study.

## 5. Conclusions

This study indicates that among the various measures of physical activity, the time spent on outdoor activities experienced the most significant decline during the COVID-19 period. This reduction in outdoor activity likely contributed to a decrease in overall physical activity levels, which in turn is a primary factor behind the notable decline in health-related physical fitness indicators, particularly in strength and endurance.

Health promoters face a significant challenge in reversing recent trends to benefit the developing generation. There is a pressing need for motivated teachers, coaches, and parents who can lead by example, encouraging children to increase their physical activity, spend more time outdoors, and engage in various sports.

## Figures and Tables

**Table 1 ijerph-21-00744-t001:** Participant’s anthropometric data: mean (M) ± standard deviation (SD), 95% confidence interval (95% CI), and number of participants (n) for both study years.

Gender and Age Groups	Weight (kg) M ± SD (n)			Height (m) M ± SD (n)			BMI (kg/m^2^) M ± SD (n)		
	2017	2022	*p*-Value	2017	2022	*p*-Value	2017	2022	*p*-Value
Females									
12–13-y	50.83 ± 10.82	50.28 ± 10.25	0.472	1.60 ± 0.07	1.61 ± 0.07	0.089	19.78 ± 3.52	19.38 ± 3.21	0.086
(95% CI)	(49.87–51.80)	(49.11–51.45)		(1.593–1.605)	(1.600–1.614)		(19.488–20.068)	(19.027–19.730)	
(n)	(581)	(395)		(594)	(412)		(580)	(394)	
14–15-y	57.69 ± 9.83	57.51 ± 10.41	0.820	1.66 ± 0.06	1.67 ± 0.07	0.036	20.91 ± 3.24	20.59 ± 3.17	0.187
(95% CI)	(56.64–58.75)	(56.30–58.72)		(1.653–1.666)	(1.663–1.678)		(20.597–21.233)	(20.224–20.953)	
(n)	(484)	(368)		(487)	(380)		(482)	(367)	
16–17-y	60.86 ± 10.06	59.83 ± 8.98	0.280	1.68 ± 0.06	1.69 ± 0.06	0.271	21.57 ± 3.20	21.06 ± 2.90	0.078
(95% CI)	(59.70–62.01)	(58.36–61.30)		(1.671–1.686)	(1.676–1.694)		(21.218–21.913)	(20.619–21.502)	
(n)	(405)	(250)		(410)	(253)		(405)	(250)	
Males									
12–13-y	51.50 ± 12.79	52.08 ± 12.59	0.451	1.60 ± 0.09	1.63 ± 0.09	<0.001	19.92 ± 3.95	19.66 ± 3.78	0.247
(95% CI)	(50.58–52.42)	(50.88–53.28)		(1.597–1.608)	(1.619–1.634)		(19.648–20.201)	(19.293–20.017)	
(n)	(641)	(373)		(646)	(396)		(638)	(372)	
14–15-y	64.25 ± 12.82	66.22 ± 14.75	0.013	1.74 ± 0.08	1.76 ± 0.08	0.018	21.09 ± 3.83	21.36 ± 4.16	0.264
(95% CI)	(63.23–65.27)	(65.04–67.40)		(1.738–1.751)	(1.749–1.764)		(20.783–21.399)	(21.004–21.714)	
(n)	(520)	(388)		(517)	(395)		(515)	(387)	
16–17-y	72.87 ± 13.95	70.15 ± 12.61	0.006	1.81 ± 0.07	1.81 ± 0.07	0.627	22.22 ± 3.72	21.53 ± 3.22	0.019
(95% CI)	(71.73–74.01)	(68.64–71.71)		(1.801–1.816)	(1.796–1.815)		(21.875–22.562)	(21.066–21.993)	
(n)	(413)	(228)		(413)	(226)		(413)	(226)	

**Table 2 ijerph-21-00744-t002:** The distribution of participants according to the frequency of exercise in different years of the study.

Gender and Age Groups	2017			2022				
	Low	Moderate	High	Low	Moderate	High	χ^2^	*p*-Value
All Students	22.0%	36.7%	41.3%	21.3%	34.4%	44.3%	4.396	0.111
(n)	(697)	(1162)	(1309)	(394)	(637)	(820)		
Females								
12–13-y	22.2%	39.4%	38.4%	24.7%	38.2%	37.2%	0.813	0.666
(n)	(136)	(241)	(235)	(97)	(150)	(146)		
14–15-y	21.0%	38.6%	40.4%	24.5%	34.4%	41.1%	2.085	0.352
(n)	(112)	(206)	(216)	(81)	(114)	(136)		
16–17-y	31.5%	35.6%	32.9%	26.9%	34.5%	38.6%	2.460	0.292
(n)	(137)	(155)	(143)	(60)	(77)	(86)		
Males								
12–13-y	17.2%	36.3%	46.2%	15.6%	33.6%	50.8%	2.051	0.359
(n)	(113)	(240)	(303)	(60)	(129)	(195)		
14–15-y	20.0%	34.2%	45.8%	17.1%	31.7%	51.2%	2.689	0.261
(n)	(106)	(181)	(243)	(63)	(117)	(189)		
16–17-y	23.2%	34.7%	42.1%	21.9%	33.1%	45.0%	0.376	0.829
(n)	(93)	(139)	(169)	(33)	(50)	(68)		

**Table 3 ijerph-21-00744-t003:** The distribution of participants according to the hours of intensive exercise during the week in different years of the study.

Gender and Age Groups	2017			2022				
	Low	Moderate	High	Low	Moderate	High	χ^2^	*p*-Value
All Students	34.8%	31.5%	33.6%	39.1%	27.4%	33.5%	12.379	0.002
(n)	(1104)	(999)	(1065)	(723)	(507)	(621)		
Females								
12–13-y	35.6%	34.3%	30.2%	46.1%	28.5%	25.4%	10.979	0.004
(n)	(217)	(209)	(184)	(181)	(112)	(100)		
14–15-y	36.0%	34.7%	29.3%	40.5%	26.9%	32.6%	5.770	0.056
(n)	(192)	(185)	(156)	(134)	(89)	(108)		
16–17-y	41.4%	28.7%	29.9%	41.3%	29.1%	29.6%	0.013	0.993
(n)	(180)	(125)	(130)	(92)	(65)	(66)		
Males								
12–13-y	32.9%	30.5%	36.6%	33.1%	27.6%	39.3%	1.169	0.557
(n)	(216)	(200)	(240)	(127)	(106)	(151)		
14–15-y	32.5%	28.0%	39.6%	37.4%	27.4%	35.2%	2.665	0.264
(n)	(173)	(149)	(211)	(138)	(101)	(130)		
16–17-y	31.4%	32.7%	35.9%	33.8%	22.5%	43.7%	5.725	0.057
(n)	(126)	(131)	(144)	(51)	(34)	(66)		

**Table 4 ijerph-21-00744-t004:** The distribution of participants according to physically active outdoor hours across different years of the study.

Gender and Age Groups	2017			2022				
	Low	Moderate	High	Low	Moderate	High	χ^2^	*p*-Value
All Students	23.5%	35.6%	40.9%	31.7%	35.4%	32.8%	49.708	<0.001
(n)	(739)	(1117)	(1285)	(588)	(656)	(608)		
Females								
12–13-y	21.4%	37.7%	40.9%	34.8%	35.0%	30.2%	23.847	<0.001
(n)	(130)	(229)	(248)	(137)	(138)	(119)		
14–15-y	24.0%	36.0%	40.0%	33.2%	40.2%	26.6%	17.813	<0.001
(n)	(127)	(191)	(212)	(110)	(133)	(88)		
16–17-y	25.3%	40.6%	34.1%	32.3%	37.7%	30.0%	3.613	0.164
(n)	(110)	(176)	(148)	(72)	(84)	(67)		
Males								
12–13-y	21.9%	33.0%	45.1%	28.1%	35.7%	36.2%	8.875	0.012
(n)	(142)	(214)	(292)	(108)	(137)	(139)		
14–15-y	26.1%	33.0%	40.8%	27.9%	33.1%	39.0%	0.430	0.807
(n)	(137)	(173)	(214)	(103)	(122)	(144)		
16–17-y	23.4%	33.7%	43.0%	38.4%	27.8%	33.8%	12.464	0.002
(n)	(93)	(134)	(171)	(58)	(42)	(51)		

**Table 5 ijerph-21-00744-t005:** Results of strength test indicators of the participants: mean (M) ± standard deviation (SD), 95% confidence interval (95% CI) for means and mean differences (M. Dif.), and number of participants (n) across different years of the study.

Gender and Age Groups	Standing Broad Jump (m) M ± SD				Bent-Arm Hang (s) M ± SD			
	2017	2022	*p*	M. Dif.	2017	2022	*p*	M. Dif.
Females								
12–13-y	1.58 ± 0.25	1.52 ± 0.30	<0.001	−0.065 (−0.030–−0.100)	9.40 ± 13.05	6.78 ± 7.43	0.003	−2.615 (−0.880–−4.351)
(95% CI)	(1.56–1.61)	(1.49–1.55)			(8.31–10.49)	(5.45–8.13)		
(n)	(591)	(391)			(587)	(384)		
14–15-y	1.66 ± 0.24	1.60 ± 0.26	<0.001	−0.063 (−0.027–−0.099)	8.79 ± 10.37	6.19 ± 9.67	0.005	−2.602 (−0.772–−4.431)
(95% CI)	(1.64–1.68)	(1.57–1.62)			(7.61–9.97)	(4.79–7.58)		
(n)	(512)	(386)			(502)	(358)		
16–17-y	1.71 ± 0.25	1.62 ± 0.29	0.001	−0.072 (−0.029–−0.115)	8.23 ± 10.04	5.61 ± 7.73	0.016	−2.624 (−0.492–−4.757)
(95% CI)	(1.69–1.74)	(1.61–1.68)			(6.91–9.56)	(3.94–7.28)		
(n)	(400)	(251)			(397)	(251)		
Males								
12–13-y	1.68 ± 0.27	1.72 ± 0.28	0.016	0.041 (0.075–0.008)	12.51 ± 13.52	12.09 ± 14.70	0.628	−0.417 (1.269–−2.103)
(95% CI)	(1.66–1.70)	(1.70–1.75)			(11.47–13.55)	(10.76–13.42)		
(n)	(647)	(419)			(647)	(397)		
14–15-y	1.97 ± 0.30	1.91 ± 0.30	0.002	−0.057 (−0.022–−0.093)	20.28 ± 17.94	17.33 ± 15.47	0.001	−2.949 (−1.192–−4.706)
(95% CI)	(1.95–1.99)	(1.89–1.94)			(19.14–21.43)	(16.00–18.67)		
(n)	(526)	(389)			(535)	(393)		
16–17-y	2.21 ± 0.26	2.16 ± 0.28	0.011	−0.057 (−0.013–−0.101)	30.03 ± 17.37	27.55 ± 17.36	0.026	−2.483 (−0.296–−4.671)
(95% CI)	(2.19–2.24)	(2.12–2.19)			(28.72–31.35)	(25.80–29.30)		
(n)	(411)	(229)			(407)	(228)		

**Table 6 ijerph-21-00744-t006:** Results of endurance indicators of the participants: mean (M) ± standard deviation (SD), 95% confidence interval (95% CI) for means and mean differences (M. Dif.), and number of participants (n) across different years of the study.

Gender and Age Groups	Curl-Up (Number of Repetition) M ± SD				20 m Shuttle Run (Number of Stages) M ± SD			
	2017	2022	*p*	M. Dif.	2017	2022	*p*	M. Dif.
Females								
12–13-y	55.53 ± 21.87	45.70 ± 22.59	<0.001	−9.824 (−6.956–−12.692)	32.74 ± 16.57	25.55 ± 13.42	<0.001	−7.198 (−4.520–−9.876)
(95% CI)	(53.74–57.32)	(43.46–47.95)			(31.11–34.37)	(23.42–27.67)		
(n)	(591)	(375)			(580)	(341)		
14–15-y	52.22 ± 22.77	45.52 ± 23.19	<0.001	−6.708 (−3.735–−9.680)	35.36 ± 19.10	25.50 ± 13.01	<0.001	−9.863 (−7.091–−12.636)
(95% CI)	(50.31–54.14)	(43.25–47.79)			(33.61–37.11)	(23.35–27.65)		
(n)	(513)	(366)			(503)	(333)		
16–17-y	51.48 ± 22.17	46.84 ± 22.62	0.009	−4.640 (−1.168–−8.113)	36.48 ± 18.10	29.82 ± 15.25	<0.001	−6.659 (−3.458–−9.860)
(95% CI)	(49.33–53.63)	(44.12–49.57)			(34.51–38.46)	(27.31–32.34)		
(n)	(408)	(254)			(394)	(243)		
Males								
12–13-y	56.61 ± 22.02	51.65 ± 21.97	<0.001	−4.964 (−2.136–−7.792)	38.86 ± 20.58	34.95 ± 17.10	0.003	−3.909 (−1.344–−6.474)
(95% CI)	(54.90–58.33)	(49.40–53.90)			(37.31–40.41)	(32.91–36.99)		
(n)	(643)	(373)			(640)	(369)		
14–15-y	59.26 ± 24.35	53.26 ± 21.73	<0.001	−5.998 (−3.076–−8.919)	51.22 ± 25.26	41.02 ± 20.94	<0.001	−10.206 (−7.540–−12.87)
(95% CI)	(57.38–61.15)	(51.03–55.50)			(49.52–52.93)	(38.97–43.07)		
(n)	(531)	(379)			(531)	(366)		
16–17-y	62.81 ± 18.02	57.57 ± 21.33	0.004	−5.240 (−1.650–−8.831)	65.54 ± 28.80	53.06 ± 22.54	<0.001	−12.473 (−9.208–−15.738)
(95% CI)	(60.66–64.96)	(54.69–60.45)			(63.59–67.48)	(50.44–55.69)		
(n)	(409)	(228)			(407)	(224)		

## Data Availability

The database of this work is at the disposal of the corresponding author and is available upon reasonable request.

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
