# Peer review of "Comparison of Estonian Schoolchildren’s Physical Activity and Fitness Indicators before and after the COVID-19 Pandemic’s Period of Restricted Mobility"

_ijerph, 2024, doi:10.3390/ijerph21060744_

Round 1

Reviewer 1 Report

Comments and Suggestions for Authors

On the whole this is a well written paper which addresses pre and post COVID-19 measures. The introduction requires you to support your statements with actual data so the reader can look at the differences being mentioned. The following amendments are required:

Abstract

Lines 10/11 – what the physical fitness indicators you are referring to?

Line 15 – 12-17 years.

Future recommendations?

Introduction

Lines 37-41 - Include data to support your statements.

Lines 44-45 - remove a one sentence/two-line paragraph.

Lines 55-56 – any data to support a decline in duration and frequency?

Line 56-58 – what did it go from to?

Lines 61-62 – data to support the increase in sedentary time?

Line 71 – add age in brackets at the end of the sentence.

Materials and Methods

Line 80 – as being at a health risk…

Line 87-88 – what specifically were the absences, provide data so the reader is fully aware.

Line 107 – can I clarify no sign in consent form was distributed and required, only a sign-out process? Please clarify?

Line 116- do you have the ethical number/code you could provide?

Results

Table 2 – have ‘moderate’ hours significantly decreased?

Table 4 – title needs amending to flow and sell the table.

Line 229 – 14-17 years

Line 230 -12-13 years

Line 223 and 234 – add ‘years’ as the unit of measure for age.

Discussion

Line 251 – study’s

Line 260 – 14-18 years

Line 295 -years

Line 329-330 – need rewording to flow

Line 336 – years

Comments on the Quality of English Language

Some sentences need rewording to flow better and make your points in a clearer manner, but on the whole the paper is well written.

Author Response

Dear Reviewer 1,

Thank you very much for reviewing the manuscript and for your suggestions for additions and changes.

Below is a description of the changes we made based on your suggestions in italics, and the corresponding places in the manuscript file are highlighted on a yellow background. Quite a few changes were made to the manuscript based on the Reviewer's 2 suggestions.

Abstract

Lines 10/11 – what the physical fitness indicators you are referring to?

* The corresponding addition has been added to line 11.

Line 15 – 12-17 years.

* Done

Future recommendations?

* I'm really sorry, but these didn't fit into the abstract because there is a word limit. Our abstract is currently exactly 200 words. We had to add a clearer conclusion there at the suggestion of Reviewer 2.

 Introduction

Lines 37-41 - Include data to support your statements.

* Done - currently on lines 46-50

Lines 44-45 - remove a one sentence/two-line paragraph.

* Done

Lines 55-56 – any data to support a decline in duration and frequency?

* Done - currently on lines 65-66

Line 56-58 – what did it go from to?

* Done - due to restrictions requiring physical distancing - currently on lines 67-68

Lines 61-62 – data to support the increase in sedentary time?

* Done - currently on lines 73-75

Line 71 – add age in brackets at the end of the sentence.

* Done - currently on line 83

Materials and Methods

Line 80 – as being at a health risk…

* The word „group“ has been added. Currently on line 97.

Line 87-88 – what specifically were the absences, provide data so the reader is fully aware.

* This sentence was suggested by Reviewer 2 to be included in the limitations of the study, added subsection, which is found in line 485 at the end of the discussion.

Line 107 – can I clarify no sign in consent form was distributed and required, only a sign-out process? Please clarify?

* Thanks for the question. We wrote this part more clearly, it can now be found on lines 138-144.

Line 116- do you have the ethical number/code you could provide?

* Done - currently on line 164

Results

Table 2 – have ‘moderate’ hours significantly decreased?

* No, it is not.

Table 4 – title needs amending to flow and sell the table.

* We reworded. Currently on line 317.

Line 229 – 14-17 years

* Done - currently on line 322

Line 230 -12-13 years

* Done - currently on line 323

Line 223 and 234 – add ‘years’ as the unit of measure for age.

* Here, the results section has been modified because a new statistical analysis was performed. Currently on lines 329-336.

Discussion

Line 251 – study’s

* Done - currently on line 366

Line 260 – 14-18 years

* Done - currently on line 375

Line 295 -years

* Done - currently on line 419

Line 329-330 – need rewording to flow

* This sentence was deleted due to text correction.

Line 336 – years

* This sentence was changed to update the results.

Comments on the Quality of English Language

Some sentences need rewording to flow better and make your points in a clearer manner, but on the whole the paper is well written.

* Thanks for the pleasant feedback. We hope that the updated manuscript is also easy to understand.

Reviewer 2 Report

Comments and Suggestions for Authors

First of all, I would like to thank you for the opportunity that was given to me to review the manuscript entitled "Comparison of Estonian Schoolchildren's Physical Activity and Fitness Indicators Before and After the COVID-19 Pandemic's Period of Restricted Mobility" which aims to determine if and to what degree the primary indicators of physical activity (exercise frequency, exercise intensity, and outdoor physical activity) and health-related physical fitness (strength and endurance) among schoolchildren have shifted, by comparing data from before and after the coronavirus pandemic period.

I consider the theme of this article to be relevant,  but I therefore make some suggestions that I believe could be a contribution to clarifying and enriching the article.

Title: In my opinion, the title is fine, clear and informative.

Abstract: Line 21. abstract: a conclusion of the study should be written more clearly

Key words: It is suggested to the authors, since this is not advisable, that the keywords that are in the title be changed to others.

Introduction.

Although the introduction is easy to read, it should be more detailed about the influence and repercussions of practicing physical activity on strength and endurance and what studies have been carried out in this regard and their results. Is it the same amount of time to practice physical activities in rural areas as in urban areas? Do boys and girls do physical activity for the same amount of time? Are the same physical activities carried out in the different age ranges studied? The authors are suggested to delve deeper into these questions and present them in the introduction.

It seems to me that the definition of objectives should be more detailed, with a main objective and well-defined secondary objectives. This definition would allow us to draw more objective conclusions and better structure the discussion of the article. It could also help the discourse in the discussion to pose research questions that can be answered throughout the discussion.

Material and methods

Line 87-88. This should be included as a limitation of the study.

Line 99: Table 1 are results so they should be transferred to this section

Line 101: The organization of the study is not clear. This can lead to confusion. On the one hand, it can be understood that the same students are evaluated, but also that they are different students. If they are the same, the statistical tests are not correct (carry out what is requested in this section) if they are different, the main problem is that the data cannot be generalized because all the strange variables that may influence have not been taken into account on it.

Line 112. The authors must indicate the following questions: How were the students measured? Who has carried out the measurements? Has a warm up been carried out? Have they been carried out at the same times in all centers, and years of measurement? In 2022 the same criteria have been followed as in 2017: the authors must explain this. Not following the same procedure could have introduced errors in the results. (include these aspects in limitations)

Statistical analyses: Statistical tests are not sufficiently described. Authors must remember that these must be clearly described so that they can be reproduced and replicated.

Lines 177-182: The statistical tests are not considered appropriate since the way they were carried out takes into account the entire sample without discriminating by gender, age or period in a general way.

Therefore, the authors are recommended to perform the following analyses: for quantitative variables, it is recommended to perform a three-factor ANOVA (time x age x gender), using time as a repeated measures factor, [i.e. Time (2017 vs 2022), age (12-13; 14-15; 16-17) and gender (male vs female)] to analyze the possible main effect of those factors on the physical condition variables and their interaction using the statistic of Bonferroni; Calculate the effect size in terms of eta squared (η2).

Additionally, the CI of the test results must be indicated.

Results:

The results section must be completely rewritten based on the requested statistical results.

Line 226: These results should be changed based on the recommended statistical tests. the confidence index and the effect size must be included

Discusion

Discussion was prepared at a very general level. At this level you can write on the Introduction part. Discussion requires a more detailed comparative analysis of research results. The discussion should confront the results of other investigations especially those quoted in the text.

Furthermore, the discussion must be changed and adapted based on the new results provided by the suggested statistical tests.

In the introduction, research questions are posed that must be answered clearly in the discussion.

Lines 308-3011. This circumstance must be included as a limitation of the study since the questionnaire or the tests were not carried out under the same circumstances and therefore the results may have been altered.

Line 362: A section on limitations of the study must be included

Conclusions

line 363. the conclusions must be modified and improved based on the new suggested statistical analyzes.

References:

20 of the 51 references (40%) are more than 5 years old. Authors are recommended to introduce more recent references, especially in the discussion section.

I hope all comments help the authors improve their manuscript.

Author Response

Dear Reviewer 2,

We are very grateful for your thorough feedback.

Below is a description of the changes we made based on your suggestions in italics, and the corresponding places in the manuscript file are highlighted on a yellow background. Some changes have been introduced based on reviewer 1's advice.

Abstract: Line 21. abstract: a conclusion of the study should be written more clearly

 * Done - currently on lines 22-23

Key words: It is suggested to the authors, since this is not advisable, that the keywords that are in the title be changed to others.

* Done - currently on line 24-25

Introduction.

Although the introduction is easy to read, it should be more detailed about the influence and repercussions of practicing physical activity on strength and endurance and what studies have been carried out in this regard and their results. Is it the same amount of time to practice physical activities in rural areas as in urban areas? Do boys and girls do physical activity for the same amount of time? Are the same physical activities carried out in the different age ranges studied? The authors are suggested to delve deeper into these questions and present them in the introduction.

* We added more specific examples there. All changes are highlighted on a yellow background.

It seems to me that the definition of objectives should be more detailed, with a main objective and well-defined secondary objectives. This definition would allow us to draw more objective conclusions and better structure the discussion of the article. It could also help the discourse in the discussion to pose research questions that can be answered throughout the discussion.

*We included specific research questions at the end of the introduction.

* Currently on lines 83-88.

Material and methods

Line 87-88. This should be included as a limitation of the study.

* Done – currently on lines 486-488

Line 99: Table 1 are results so they should be transferred to this section

* Done – currently on line 274

 Line 101: The organization of the study is not clear. This can lead to confusion. On the one hand, it can be understood that the same students are evaluated, but also that they are different students. If they are the same, the statistical tests are not correct (carry out what is requested in this section) if they are different, the main problem is that the data cannot be generalized because all the strange variables that may influence have not been taken into account on it.

* We supplemented the subsection "Organization of the study", which begins on line 133.

* And we tried to better explain the inclusion of different students in different years of the study right in the introductory part of the "Materials and Methods" chapter, on lines 99-101 and 104-106.

Line 112. The authors must indicate the following questions: How were the students measured? Who has carried out the measurements? Has a warm up been carried out? Have they been carried out at the same times in all centers, and years of measurement? In 2022 the same criteria have been followed as in 2017: the authors must explain this. Not following the same procedure could have introduced errors in the results. (include these aspects in limitations)

* A more detailed explanation of how to conduct physical fitness tests is on the lines 150-156 and the more precise methodology is outlined in the subsection "Physical fitness tests", which starts from the line 214.

Statistical analyses: Statistical tests are not sufficiently described. Authors must remember that these must be clearly described so that they can be reproduced and replicated.

Lines 177-182: The statistical tests are not considered appropriate since the way they were carried out takes into account the entire sample without discriminating by gender, age or period in a general way.

Therefore, the authors are recommended to perform the following analyses: for quantitative variables, it is recommended to perform a three-factor ANOVA (time x age x gender), using time as a repeated measures factor, [i.e. Time (2017 vs 2022), age (12-13; 14-15; 16-17) and gender (male vs female)] to analyze the possible main effect of those factors on the physical condition variables and their interaction using the statistic of Bonferroni; Calculate the effect size in terms of eta squared (η2).

Additionally, the CI of the test results must be indicated.

* Thanks for the suggestion. We performed the appropriate ANOVA test and also rewrote the subsection "Statistical analysis"", which starts from the line 257.

Results:

The results section must be completely rewritten based on the requested statistical results.

Line 226: These results should be changed based on the recommended statistical tests. the confidence index and the effect size must be included

* Done. New results start at line 319.

Discusion

Discussion was prepared at a very general level. At this level you can write on the Introduction part. Discussion requires a more detailed comparative analysis of research results. The discussion should confront the results of other investigations especially those quoted in the text.

Furthermore, the discussion must be changed and adapted based on the new results provided by the suggested statistical tests.

In the introduction, research questions are posed that must be answered clearly in the discussion.

* We made quite a few changes in the "discussion" chapter. All changed parts are highlighted with a yellow background.

* When comparing the results of physical fitness, the authors did not consider it necessary to bring out specific numbers, as changes over time were primarily evaluated.

Lines 308-3011. This circumstance must be included as a limitation of the study since the questionnaire or the tests were not carried out under the same circumstances and therefore the results may have been altered.

* Done – currently on lines 489-493

Line 362: A section on limitations of the study must be included

* "Limitations of the study" begins on line 485.

Conclusions

line 363. the conclusions must be modified and improved based on the new suggested statistical analyzes.

* Done -  currently on lines 498-502

References:

20 of the 51 references (40%) are more than 5 years old. Authors are recommended to introduce more recent references, especially in the discussion section.

* Thank you for your attention. We updated sources where possible.

Round 2

Reviewer 2 Report

Comments and Suggestions for Authors

The authors have made the changes suggested by me and have improved the manuscript so that it can be published in its current form.